# Gene Expression Changes in Cytokine and Chemokine Receptors in Association with Melanoma Liver Metastasis

**DOI:** 10.3390/ijms24108901

**Published:** 2023-05-17

**Authors:** Viktória Koroknai, István Szász, Margit Balázs

**Affiliations:** 1Department of Public Health and Epidemiology, Faculty of Medicine, University of Debrecen, 4032 Debrecen, Hungary; koroknai.viktoria@med.unideb.hu (V.K.); szasz.istvan@med.unideb.hu (I.S.); 2ELKH-DE Public Health Research Group, University of Debrecen, 4032 Debrecen, Hungary

**Keywords:** malignant melanoma, liver metastasis, cytokines, *IL11RA*, hepatic endothelial cells

## Abstract

Cytokines and chemokines (chemotactic cytokines) are soluble extracellular proteins that bind to specific receptors and play an integral role in the cell-to-cell signaling network. In addition, they can promote the homing of cancer cells into different organs. We investigated the potential relationship between human hepatic sinusoidal endothelial cells (HHSECs) and several melanoma cell lines for the expression of chemokine and cytokine ligands and receptor expression during the invasion of melanoma cells. In order to identify differences in gene expression related to invasion, we selected invasive and non-invasive subpopulations of cells after co-culturing with HHSECs and identified the gene expression patterns of 88 chemokine/cytokine receptors in all cell lines. Cell lines with stable invasiveness and cell lines with increased invasiveness displayed distinct profiles of receptor genes. Cell lines with increased invasive capacity after culturing with conditioned medium showed a set of receptor genes (*CXCR1*, *IL1RL1*, *IL1RN*, *IL3RA*, *IL8RA*, *IL11RA*, *IL15RA*, *IL17RC*, and *IL17RD*) with significantly different expressions. It is very important to emphasize that we detected significantly higher *IL11RA* gene expression in primary melanoma tissues with liver metastasis as well, compared to those without metastasis. In addition, we assessed protein expression in endothelial cells before and after co-culturing them with melanoma cell lines by applying chemokine and cytokine proteome arrays. This analysis revealed 15 differentially expressed proteins (including CD31, VCAM-1, ANGPT2, CXCL8, and CCL20) in the hepatic endothelial cells after co-culture with melanoma cells. Our results clearly indicate the interaction between liver endothelial and melanoma cells. Furthermore, we assume that overexpression of the *IL11RA* gene may play a key role in organ-specific metastasis of primary melanoma cells to the liver.

## 1. Introduction

Efficient communication between cells is one of the most important aspects of cell conditions in multicellular organisms [1]. Metastasis formation in primary tumors is also a balance of the host (‘soil’) and the tumor (‘seed’) cellular interactions, an altered cross-talk between the cells whose mechanism is determined by both the intrinsic properties of the tumor cells and the host response [2,3,4]. According to the ‘seed and soil hypothesis’, the primary tumor contributes to the preparation of secondary sites for tumor cell invasion and development, known as the pre-metastatic niche (PMN) [5,6]. The distribution of the targeted organ is especially variable depending on the cancer type; however, melanoma has the unique property of having a fully comprehensive metastatic potential; hence, any organ or tissue can host melanoma metastasis [7,8]. Metastasis is the main factor limiting survival in most patients with cancer. Therefore, understanding the molecular mechanisms that control the metastatic behavior of tumor cells is an important key to the successful treatment of cancer.

Cytokines and chemokines (chemotactic cytokines) are soluble extracellular proteins that are an integral part of the cell-to-cell signaling network and can bind to specific receptors to control cellular growth, development, hematopoiesis, lymphocyte recruitment, inflammation, and immune regulation [9]. There are numerous observations that endothelial cells coating the blood vessels of various organs express a number of different cytokines, chemokines, and adhesion molecules that bind to specific receptors on the cell surface to promote the homing of cancer cells [10,11,12]. In this way, the site of secondary tumor formation depends on the chemokine receptors expressed on the tumor cell surface and the chemokine expression specific to each tissue. The use of cytokines in the clinic as monotherapy due to their need for high and frequent dosing often results in toxicity [13]. Combinations of cytokines with various checkpoint inhibitors may provide promising aspects for tumor treatment, including melanoma [14].

The most common sites of distant metastases in melanoma patients include the skin, liver, lung, and brain; however, the initial sites of melanoma cell spread are typically lymph nodes [7,15]. Liver metastases are observed in 10–20% of patients with cutaneous melanoma and occur relatively late during disease progression, with an average of 2–4 months of survival [15,16,17]. The liver has a rich blood supply and therefore provides fertile “soil” for metastatic spread [18]. Liver sinusoidal endothelial cells (LSECs) constitute approximately 50% of the non-parenchymal cells of the liver and form the fenestrated wall of the hepatic sinusoids with minimal basement membrane [19,20]. LSECs are the first cells to contact the blood flow in hepatic sinusoids, and they have several pro-metastatic properties [21]. LSECs express numerous cellular adhesion molecules, including ICAM-1, VCAM-1, endothelial (E)-selectin, and CD31, which can facilitate cancer cell migration and promote the tumor cells adherence to the LSECs, extravasation, and metastasis formation [22]. Mendt et al. have described the possible role of the CXCL12/CXCR4 axis in liver metastasis of melanoma; however, little is known about the specific factors that regulate the growth of liver metastases [23]. Importantly, different cytokines and chemokines are also expressed by the LSECs, which play an important role in recruiting distinct leukocytes to contribute to the immune microenvironment within the liver [21,24].

Different chemokines and cytokines secreted by endothelial cells may have different effects on the invasion properties of melanoma cells expressing specific receptors. Therefore, we aimed to investigate the potential relationship between human hepatic sinusoidal endothelial cells (HHSECs) and several different melanoma cell lines in terms of chemokine and cytokine ligand and receptor expression during in vitro invasion of melanoma cells. We hypothesize that the study of the ligand/receptor axis expressed during melanoma cell invasion targeting hepatic endothelial cells may provide new insights into the understanding of the mechanism of liver-specific metastasis of melanoma.

## 2. Results

### 2.1. Effect of HHSEC on Melanoma Cell Invasion

To investigate the impact of HHSEC cells on melanoma cell invasion, an invasion assay was implemented using human hepatic sinus endothelial cells in conditioned media (HHSEC-CM) as a chemoattractant or unconditioned media containing 10% fetal bovine serum (FBS) as a control (Appendix A). We observed that all cell lines showed invasive properties; however, significantly increased cell invasion was detected in three cases (WM983A, WM3248, and WM1360) after co-culturing melanoma cells with HHSEC-CM compared to culturing in unconditioned medium (Figure 1). Cell lines (WM1366, WM278, and WM793B) that did not change their invasive properties regardless of the medium (conditioned or not) were referred to as stable invasive.

### 2.2. Chemokine and Cytokine Receptor Expression of Melanoma Cells

In order to define the gene expression differences related to melanoma cell invasion, we selected the invasive and non-invasive cells after co-culturing with HHSEC in every cell line. In this way, we were able to establish selected invasive cell populations (WM1366-H^INV^, WM278-H^INV^, WM793B-H^INV^, WM983A-H^INV^, WM1361-H^INV^, and WM3248-H^INV^; H = hepatic and INV = invasive) and selected non-invasive cell populations (WM1366-H^NON-INV^, WM278-H^NON-INV^, WM793B-H^NON-INV^, WM983A-H^NON-INV^, WM1361-H^NON-INV^, and WM3248-H^NON-INV^; H = hepatic and NON-INV = non-invasive). After the selection, we determined the gene expression patterns of six invasive and six non-invasive cell populations (Appendix A).

To investigate the gene-expression differences triggered by the hepatic endothelial cells in response to the HHSEC-conditioned medium, two main groups were generated: cell lines with stable invasiveness (WM1366, WM278, and WM793B) and cell lines with increased invasiveness (WM983A, WM1361, and WM3248). After that, we analyzed the expression changes between the selected invasive and non-invasive cells in the two groups. Melanoma cell lines with stable invasive properties exhibited a small cohort of genes (*XCR1*, *IL1RN*, *IL11RA*, *TNFRSF4*, *TNFRSF12A*, and *TNFRSF13C*) that were significantly altered in the separated invasive cells compared to the non-invasive ones (Figure 2A(a)). The same comparison was made for cell lines that responded to the conditioned medium with increased invasion, and a different group of altered genes (*CXCR1*, *IL1RL1*, *IL1RN*, *IL3RA*, *IL8RA*, *IL11RA*, *IL15RA*, *IL17RC*, and *IL17RD*) were found in the separated invasive cells (Figure 2A(b)). *IL11RA* was the only gene in the overlapping region of the two sets of gene expression results. Unsupervised hierarchical clustering of the significantly differentially expressed genes in melanoma cell lines with increased invasiveness is displayed in Figure 2B. The separated non-invasive (NON-INV) and invasive (INV) cells are displayed vertically, and the genes are displayed horizontally. The heat map clearly shows the differences between genes associated with the two subgroups within the cell lines with increased invasiveness. Based on this analysis, the *XCR1*, *IL1RN*, *TNFRSF4*, *TNFRSF12A*, and *TNFRSF13C* genes were excluded from further investigations as these gene expression changes are probably not associated with HHSEC-CM-induced invasion. At the same time, in the case of *IL11RA* (Figure 2C), the difference between the invasive and non-invasive subgroups is remarkable only in cell lines with increased invasion, while the difference in *IL11RA* gene expression in cell lines with stable invasion is marginal. These results indicate that upregulation of the *IL11RA* gene was triggered by HHSEC-CM.

### 2.3. Correlation between Gene Expression and Invasiveness

We analyzed the correlation between invasive potential and altered expression of the receptor genes resulting from co-cultivation with HHSEC-CM. We correlated the invasion changes with the ratio of gene expression data between invasive and non-invasive populations after culturing the cells with HHSEC-CM. Significantly correlated genes included four chemokine receptors, two tumor necrosis factor receptors, and seven interleukin receptor genes (Appendix A). Interestingly, we found a significant positive correlation between the expression of *IL15RA*, *IL17RC*, and *IL17RD* genes and the invasiveness of cell lines (R = 0.933, *p* = 0.007; R = 0.872, *p* = 0.024; and R = 0.938, *p* = 0.006, respectively, Figure 3), and those three receptor genes displayed significantly increased expression in the selected invasive melanoma cells after culturing with HHSEC-CM as well.

### 2.4. Proteome Profile of HHSECs

Proteome arrays (Proteome Profiler Human Chemokine Array Kit and Proteome Profiler Human XL Cytokine Array Kit) were used to detect protein expression differences between HHSEC before (HHSEC-Control) and after co-culturing with different melanoma-derived primary cell lines (HHSEC + WM1366, HHSEC + WM278, HHSEC + WM793B, HHSEC + WM983A, HHSEC + WM1361, and HHSEC + WM3248). Proteins with detectable differences (>10%) in at least one cell line revealed 15 differentially expressed proteins (including CD31, VCAM-1, ANGPT2, CXCL8, and CCL20) in the co-cultured HHSECs compared to the control (Figure 4). We observed remarkably higher expression of the CCL20 protein in HHSECs co-cultured with WM3248, suggesting a potential correlation between CCL20 expression and the invasive potential of WM3248 cells.

### 2.5. Gene Expression of Chemokine and Cytokine Receptors in Melanoma Tissue Samples

We were able to analyze the gene expression differences of chemokine- and cytokine-receptor genes for a small cohort of primary melanoma tissue samples (non-metastatic, *n* = 4; primary melanoma with liver metastasis, *n* = 5). We examined the expression of genes (*CXCR1*, *IL1RL1*, *IL1RN*, *IL3RA*, *IL8RA*, *IL11RA*, *IL15RA*, *IL17RC*, and *IL17RD*) that showed significant differences between invasive and non-invasive cells in the cell lines with increased invasiveness after culturing with HHSEC-CM. According to our qRT-PCR data, the relative expression level of the *IL11RA* gene was significantly higher in primary melanoma tissues with liver metastasis compared to primary melanoma without metastasis (Figure 5A). On the other hand, we analyzed the association of the gene expression of these receptor genes with the Breslow thickness (mm) of primary melanoma tissues. This analysis revealed a significant positive correlation for *IL17RD* gene expression (R = 0.777, *p* = 0.014; Figure 5B).

## 3. Discussion

The mechanisms of melanoma metastasis have been the subject of extensive studies for decades, as melanoma has one of the highest metastatic potentials among human cancers with diverse secondary tumor sites, including the liver [15,26,27]. However, most of the recent studies are focusing on the most common primary cancers that metastasize to the liver, including colorectal and pancreatic cancers as well as uveal melanoma [16,28,29,30]. The organotropism of the liver is influenced by different factors such as blood flow pattern, tumor stage, and histological subtype of the tumor [18]. On the other hand, it is well established that targeted organs of future metastasis do not passively participate in this process but are modified by the primary tumor via intercellular communication [5,31]. Several studies have shown that a wide range of ligands and receptors, including integrins, selectins, adherence molecules and immunoglobulin superfamily receptors, contribute to heterotypic adhesion processes between tumor cells and endothelial cells [32,33,34]. In a model of liver metastasis, it was shown that the CXCR4 chemokine receptor on tumor cells and its ligand CXCL12 expressed by endothelial cells significantly promote tumor cell transendothelial migration [35]. Mendt et al. also highlighted the role of CXCR4/CXCL12 in in vitro migration, invasion, proliferation, and adhesion of melanoma cells during liver metastasis [23].

We aimed to investigate the possible interaction and communication between human hepatic sinusoidal endothelial cells (HHSECs) and melanoma cells and analyze the effect of HHSEC on the invasion of the different primary tumor-derived melanoma cell lines. As it was expected, not all of the cell lines had the same response during co-culturing with HHSEC; three cell lines showed significantly increased invasive potential compared to the invasiveness before the co-culture. After co-culturing, we investigated the gene expression differences between the invasive and non-invasive melanoma cells and examined the protein expression changes of the HHSEC cells before and after the co-culturing as well.

In good concordance with previous findings, we observed expression changes of several cellular adhesion molecules, cytokines, and chemokines, including VCAM-1, VEGF, CD105, or VCAM-1, in the HHSEC cells after interacting with melanoma cells [22]. CD105 (Endoglin), which is predominantly expressed by endothelial cells and is involved in tumor angiogenesis, showed higher protein expression in HHSEC cells in four cell lines after co-culture [36]. Li et al. have interpreted that CD105 promotes the invasion and metastases of liver cancer cells by increasing VEGF expression [37]. Vascular cell adhesion molecule-1 (VCAM-1) was also overexpressed in HHSEC as a result of co-culturing in three melanoma cell lines. Previously, it was described that IL-18 is able to increase the expression of VCAM-1 in the hepatic sinusoidal epithelium, promoting the adherence of melanoma cells [38]. Blocking IL-18 with a soluble factor can decrease the adhesion of melanoma cells by inhibiting this mechanism [39]. On the other hand, increased protein expression of Angiopoietin-2 (ANGPT2) was also observed in HHSEC after co-culturing in all of the cell lines. ANGPT2 is a nearly endothelial cell-specific cytokine that promotes vascular remodeling in an autocrine pathway, and its circulating level has been reported to be in association with the progression of metastasis in melanoma patients, particularly in stage III/IV melanoma patients [40,41,42]. Furthermore, Urosevic et al. suggested that ANGPT2 is important for metastatic outgrowth in the liver in colorectal cancer by binding to the TIE2 receptor [43].

Interleukins, such as IL-2, IL-10, IL-12, IL15, and IL-21, and their receptors have been shown to be efficient mediators of anti-tumor immunity in preclinical cancer models [44,45,46,47,48]. However, our aim was to find specific receptors that have a possible role in invasion toward hepatic endothelial cells, therefore possibly promoting the liver metastasis of melanoma cells, a mechanism that has not been well studied yet. According to our results, the conditioned medium produced by hepatic endothelial cells had a different effect on the invasion of melanoma cells. Half of the cell lines (WM983A, WM1361, and WM3248) showed increased invasion properties after co-culturing with HHSEC-CM compared to cells grown in the unconditioned medium as a control, while three cell lines (WM1366, WM278, and WM793B) had stable invasiveness compared to the controls. In all cell lines, we selected the invasive melanoma cells from the non-invasive cells and compared the chemokine and cytokine receptor gene expression of the two separated subpopulations based on their response to HHSEC-conditioned media. A set of receptor genes (*CXCR1*, *IL1RL1*, *IL1RN*, *IL3RA*, *IL8RA*, *IL11RA*, *IL15RA*, *IL17RC*, and *IL17RD*) with significantly different expression patterns was identified in cell lines with increased invasive capacity. Distinct expression patterns were observed in invasive cells compared to non-invasive cells in cell lines with stable invasiveness; genes including *XCR1*, *IL1RN*, *TNFRSF4*, *TNFRSF12A*, and *TNFRSF13C* are likely not specific for HHSEC-induced invasion. However, the *IL11RA* gene, which was included in both comparisons, was considered to belong to the group of cell lines with increased invasiveness as the difference in gene expression between invasive and non-invasive cells was hardly detectable in the cell lines with stable invasiveness (Figure 2C).

On the other hand, we examined the cytokine and chemokine expression of the hepatic endothelial cells and found 14 cytokines (CD105, CD147, CXCL8, EGF, CXCL16, Midkine, CD26, CXCL1, CD31, VEGF, CCL20, ANGPT2, VCAM-1, and TSP-1) that showed at least a 10% difference after the co-culture with melanoma cells compared to the original protein profile of the hepatic endothelial cells.

CXCR1 (C-X-C chemokine receptor 1) interacting with CXCL8 (Interleukin-8, IL-8) ligand has a well-known role in the initiation and development of various cancers, including melanoma [49,50,51]. Several studies have indicated that the upregulated *CXCR1* is associated with enhanced proliferation and invasiveness in melanoma cells; moreover, it is suggested that IL-8 and its receptor promote liver metastasis in colorectal cancer [52,53,54]. In agreement with these studies, we found that the relative mRNA level of the *CXCR1* gene was significantly higher in the selected invasive cells compared to the non-invasive ones, especially in melanoma cells with increased invasiveness after being cultured with conditioned medium from HHSEC. On the other hand, CXCL8 showed a variable expression pattern with no strong association with melanoma cell invasiveness; however, the highest expression was found in HHSEC cells co-cultured with the WM3248 cell line, which had the highest increased invasiveness of all cell lines.

In the case of the *IL1RL1* gene, which codes a specific receptor (ST2) for IL-33 interleukin, it was reported that the IL-33/ST2 axis increases the migration and invasion of melanoma cells through ERK1/2 signaling [55]. Interestingly, Luo et al. have found that IL-33/ST2 inhibits colon cancer growth and metastasis to the lung and liver; on the other hand, Zhang et al. described that increased expression of IL-33 in colorectal cancer cells enhances tumor growth and liver metastasis [56,57]. In our experiments, we found higher expression levels of the *IL1RL1* gene in melanoma cell lines with increased invasion after HHSEC-CM treatment, but we have not observed any IL-33 expression in the hepatic endothelial cells. The antitumor function of IL-33 is controversial since Gao et al. suggested that IL-33 accelerates tumor growth in the absence of lymphocytes and inhibits tumor growth by IL-33-driven immune responses [58].

The role of IL-11 expression was first studied in primary breast cancer, which developed metastasis into the bone [59,60]. IL-11 and its receptor, interleukin-11 receptor α (IL-11Rα) activate STAT3 signaling, which correlates with poor patient prognosis in most human cancers [61,62,63]. In addition, Yamazumi et al. described that the expression of IL11Rα significantly correlates with colorectal carcinoma invasion [64]. The overexpression of the *IL11RA* gene in association with melanoma liver metastasis has not been described yet, but it is important to note that we found significantly higher mRNA levels in the selected invasive melanoma cells after HHSEC-CM treatment. We had the opportunity to compare the *IL11RA* expression levels of primary melanoma tissues with and without liver metastasis; similar to our model experiments, we found a significant increase in the *IL11RA* gene expression in primary melanoma tissues with liver metastasis. Unfortunately, access to primary melanoma samples is highly limited, as indicated by the small sample size of our analysis. The other limitation of this study is the lack of comparison with other types of endothelial cells, which could give a comprehensive understanding of the organ-specific invasion strategies of melanoma cells. According to our results, we assume that our results indicate that the upregulation of the *IL11RA* gene in the invasive melanoma cells is triggered by HHSECs as well as in primary melanoma samples with liver metastasis, suggesting that *IL11RA* may play a possible role in the liver metastasis of primary melanoma cells; however, further investigations are necessary to test this hypothesis.

## 4. Materials and Methods

### 4.1. Cell Lines and Culturing

Primary tumor-derived melanoma cell lines were obtained from the Coriell Institute for Medical Research (Camden, NJ, USA). The characteristics and origins of the cell lines are summarized in Table 1. The cells were cultured in RPMI 1640 medium (Lonza Group Ltd., Basel, Switzerland) supplemented with 10% fetal bovine serum (Gibco, Carlsbad, CA, USA) at 37 °C in 5% CO_2_.

Human hepatic sinusoidal endothelial cells (HHSEC) were obtained from ScienCell Research Laboratories, Inc. (Carlsbad, CA, USA) and cultured in a 37 °C, 5% CO_2_ incubator according to the recommended protocol. HHSEC cells were maintained using Endothelial Cell Medium in a 2 μg/cm^2^ fibronectin-coated culture vessel (ScienCell Research Laboratories, Inc., Carlsbad, CA, USA). Conditioned media were collected as described before [23]. Briefly, after the endothelial cells were grown to 90% confluence, the culture medium was replaced with fresh medium and incubated for 48 h. The conditioned medium was centrifuged at 2000× *g* for 15 min and passed through a 0.2 mm filter.

### 4.2. Melanoma Tissue Samples Used for qRT-PCR

Fresh/frozen melanoma tissues were obtained from the Department of Dermatology at the University of Debrecen (Debrecen, Hungary) from patients who did not undergo therapy before the surgical removal of their primary lesions. This study was approved by the Regional and Institutional Ethics Committee of the University of Debrecen [document no.: 25364-1/2012/EKU (449/PI/12)] and carried out according to all relevant regulations. Lesions were diagnosed on the basis of formalin-fixed paraffin-embedded tissue sections stained with hematoxylineosin. A total of nine primary melanoma samples were used for the qRT-PCR. Clinical-pathological parameters of the tumors are summarized in Table 2.

### 4.3. In Vitro Invasion Assay

The invasive potential of melanoma cells was observed using BD Biocoat Matrigel invasion chambers (pore size: 8 μm, 24 wells; BD Biosciences, Bedford, MA, USA). In the control experiments, the upper chamber was filled with 500 μL of the cell suspension in serum-free media (5 × 10^4^ cells/well), and 750 μL of medium containing 10% FBS was applied to the lower chamber as a chemoattractant. To examine the effect of HHSEC on the invasion of melanoma cells, the lower chamber was filled with 750 μL of HHSEC-conditioned medium (HHSEC-CM), as described before [23]. After the cells were incubated for 24 h at 37 °C, the cells in the lower layer were fixed and stained. The invaded cells were counted using a light microscope in seven different visual fields at 200× magnification; the data are presented as the means ± SD of three independent experiments.

### 4.4. Co-Culturing of Melanoma Cell Lines and Endothelial Cells

For co-culturing, BD Biocoat Matrigel invasion chambers (pore size: 8 μm, 6 wells; BD Biosciences, Bedford, MA, USA) were used. Melanoma cells (2 × 10^5^ cells/well) were placed in the upper chamber and carefully inserted into six-well plates containing the HHSEC monolayer. Cells were incubated together for 24 h, then the insert containing melanoma cells was moved to a new plate. The invading melanoma cells and the non-invasive cells were treated separately with a 0.5% trypsin/0.2% EDTA solution (Sigma-Aldrich Inc., St. Louis, MO, USA) for recovery from the membrane and cultured using standard protocols.

### 4.5. Real-Time Quantitative PCR Analysis

RNA isolation was performed using a RNeasy Plus Mini Kit (Qiagen GmbH, Hilden, Germany) according to the manufacturer’s protocol. RNA concentration and quality were assessed using a NanoDrop (Agilent Technologies, Palo Alto, CA, USA). RNA samples with a 260/280 ratio ≥1.8 were included in further analyses. Reverse transcription of total RNA (1000 ng) was performed using a High-Capacity cDNA Archive Kit (Applied Biosystems, Carlsbad, CA, USA) according to the manufacturer’s protocol. The relative expression levels of 96 genes (20 chemokine receptor genes, 43 interleukin receptor genes, 25 tumor necrosis factor receptor genes, and 8 housekeeping genes) were determined using a LightCycler^®^ 480 Real-Time PCR System (Roche Diagnostics, GmbH, Mannheim, Germany). Primer sets (Human Cytokine and Chemokine Receptor Primer Library, RealTimePrimers.com, accessed on 11 March 2023, Elkins Park, PA, USA) were performed as described before [65]. The data are presented as the mean of 2^–ΔCt^ values from three independent experiments.

### 4.6. Protein Expression Analysis

Protein expression analyses of HHSEC cells were performed before and after co-culturing with different melanoma cell lines (WM983A, WM793B, WM1366, WM1361, and WM278), as described in detail before [66]. RIPA Lysis and Extraction Buffer (Thermo Fisher Scientific Inc., Waltham, MA, USA) containing 20 µL protease and phosphatase inhibitor cocktail (Thermo Fisher Scientific Inc., Waltham, MA, USA) was used for extraction, and the protein concentration was determined using the Pierce™ Coomassie (Bradford) Protein Assay Kit (Thermo Fisher Scientific Inc., Waltham, MA, USA). The Proteome Profiler Human Chemokine Array Kit was used to determine the expression of 31 different chemokines, and the Proteome Profiler Human XL Cytokine Array Kit was used to analyze 105 cytokines simultaneously (R&D Systems, Inc., Minneapolis, MN, USA). All the necessary reagents and the array procedure were performed according to the manufacturer’s detailed protocol. The labeled proteins were detected and visualized by the Azure c300 Chemiluminescent Imaging System (Dublin, CA, USA) using Chemi Reagent Mix (R&D Systems Inc., Minneapolis, MN, USA). Data were analyzed with AzureSpot (version: 2.2.167) software. The intensity of the positive control (reference spot) was considered 100%.

### 4.7. Statistical Analysis

IBM SPSS Statistics 26 (IBM Corp., Armonk, NY, USA) software was used for the statistical analyses. The Shapiro–Wilk test was used to evaluate the normality of the data. The Spearman’s and Pearson’s correlation coefficients were calculated to correlate the qPCR data with the invasive capacity of the melanoma cells and with the Breslow thickness of primary melanoma samples. The Mann–Whitney–Wilcoxon test and Wilcoxon signed-rank test were used to compare the qPCR data. *p* < 0.05 was considered to be statistically significant.

## 5. Conclusions

We found several potential genes and proteins with altered expression in melanoma cells that were co-cultured with human hepatic sinusoidal endothelial cells. Our findings clearly indicate the interaction between liver endothelial and melanoma cells. Based on our data, overexpression of the *IL11RA* gene might have a key role in the formation of organ-specific metastasis to the liver induced by primary melanoma cells.

## Figures and Tables

**Figure 1 ijms-24-08901-f001:**
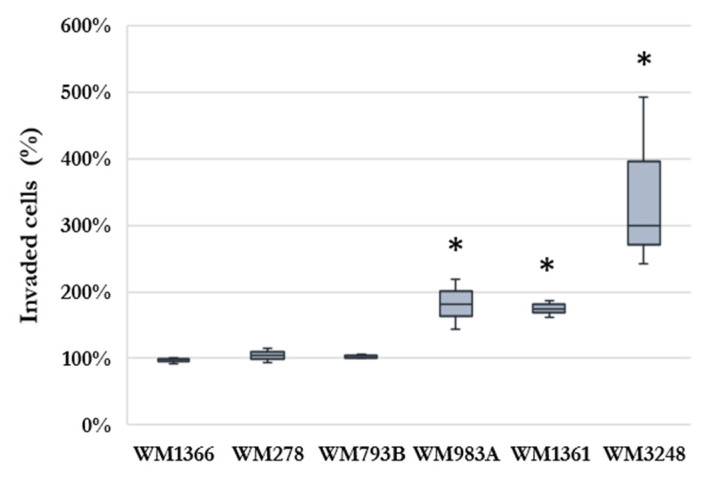
The effect of human hepatic sinusoidal endothelial cells conditioned media (HHSEC-CM) on the invasion of melanoma cell lines (WM1366, WM278, WM793B, WM983A, WM1361, and WM3248) compared to the effect of the unconditioned medium as a control. Results are presented as the ratio (mean ± SE, *n* = 3) of HHSEC-CM-induced and control invaded cells. The asterisk indicates a statistically significant difference between HHSEC-CM induced data and control data (Mann–Whitney test: * *p* < 0.05). The results of the invasion assay are shown as the mean of three independent experiments.

**Figure 2 ijms-24-08901-f002:**
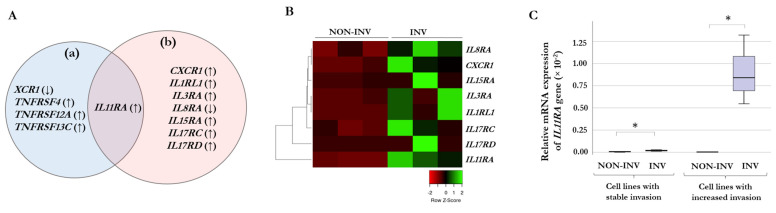
Association between chemokine- and cytokine-receptor expression of melanoma cells and the invasive capacity responding to human hepatic sinusoidal endothelial cells conditioned media (HHSEC-CM). (**A**) The Venn diagram illustrates the significant gene expression alterations in cell lines with stable invasiveness (**a**) and in those with increased invasiveness (**b**), indicating the IL11RA gene in the overlapping region of the two sets of gene expression. Arrows indicate overexpression (↑) and downregulation (↓) in the selected invasive cells compared to the non-invasive populations. (**B**) Unsupervised hierarchical clustering of the significantly differentially expressed genes in the melanoma cell lines with increased invasiveness. The separated non-invasive (NON-INV) and invasive (INV) cells are displayed vertically, and the genes are displayed horizontally. The heat map was generated from the seven significantly expressed genes using www.heatmapper.ca [25] accessed on 4 May 2023. (**C**) Relative mRNA expressions of the IL11RA gene in selected non-invasive and invasive cells according to the invasive capacity of the cell lines. The cell lines that exhibited stable invasiveness (left part) showed nearly imperceptible differences, while the cell lines with increased invasiveness (right part) demonstrated a significantly higher level of IL11RA. The data are presented as the mean ± SD of three independent experiments. An asterisk indicates statistically significant differences (Mann–Whitney–Wilcoxon test, * *p* < 0.05); the qRT-PCR results are presented as the mean ratio of 2^−ΔCt^ values between invasive and non-invasive populations of three independent experiments.

**Figure 3 ijms-24-08901-f003:**
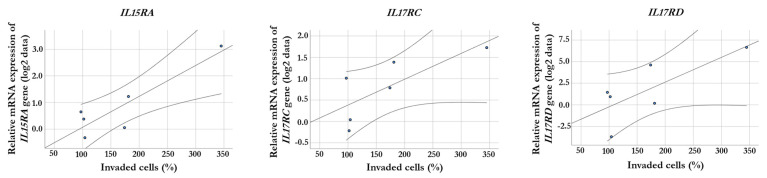
Correlation between gene expression and invasiveness in melanoma cell lines. There are significant positive correlations (*p* < 0.05) between the relative expression of the *IL15RA*, *IL17RC*, and *IL17RD* genes and the HHSEC-CM-induced invasion changes (%). The qRT-PCR results are presented as the mean ratio of 2^−ΔCt^ values between invasive and non-invasive populations (log_2_-transformed data) of three independent experiments.

**Figure 4 ijms-24-08901-f004:**
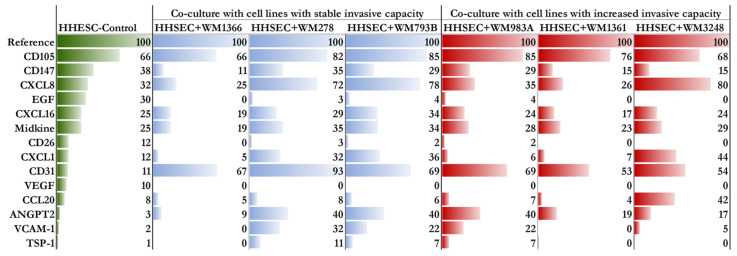
Relative protein expression profiles of human hepatic sinusoidal endothelial cells (HHSECs) co-cultured with different melanoma cell lines. The intensity of the reference is displayed at 100%. Numbers beside the columns indicate the protein expression as a percentage of the intensity of the reference spots on the array. Protein expression in control HHSECs is indicated in green; protein expression in HHSECs co-cultured with melanoma cell lines with stable invasiveness is indicated in blue; and protein expression in HHSECs co-cultured with cell lines with increased invasiveness is indicated in red.

**Figure 5 ijms-24-08901-f005:**
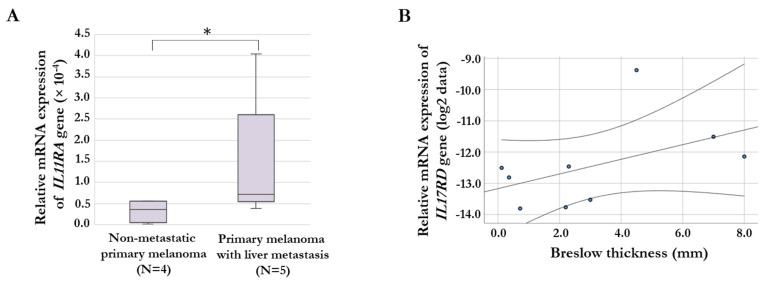
Association between cytokine receptor gene expression in primary melanoma tumor samples and clinical pathological characteristics. (**A**) Significantly increased relative expression level of the *IL11RA* gene in primary melanoma samples with liver metastasis compared to non-metastatic primary melanoma samples. An asterisk indicates a significant difference (* *p* = 0.049). (**B**) Significantly positive correlation between *IL17RD* gene expression and Breslow thickness (mm) of the primary tumors (*p* = 0.014). The qRT-PCR results are presented as 2^−ΔCt^ values.

**Table 1 ijms-24-08901-t001:** Characteristics of primary tumor-derived melanoma cell lines.

Cell Line	Growth Phase ^1^	Histologic Subtype ^2^	BRAF Mutation Status ^3^	NRAS Mutation Status ^4^
WM793B	RGP/VGP	SSM	V600E	wt
WM1361	VGP	SSM	wt	Q61L
WM278	VGP	NM	V600E	wt
WM983A	VGP	n.a.	V600E	wt
WM1366	VGP	n.a.	wt	Q61L
WM3248	VGP	n.a.	V600E	wt

^1^ RGP: radial growth phase; VGP: vertical growth phase; ^2^ SSM: superficial spreading melanoma; NM: nodular melanoma; n.a.: data not available; ^3^ V: valine; E: glutamic acid; wt: wild-type; ^4^ Q: glutamine; and L: leucine.

**Table 2 ijms-24-08901-t002:** Clinical-pathological data of melanoma tissue samples used in qRT-PCR analysis.

Sample Number	Gender ^1^	Age at Initial Diagnosis (Years)	Location	Histological Subtype ^2^	Breslow Thickness (mm)	Ulceration
Primary melanoma with no metastasis ^3^						
1	F	64	Extremities	SSM	0.4	No
2	M	67	Head	NM	0.1	No
3	M	72	Trunk	NM	4.5	No
4	M	59	Trunk	SSM	0.7	No
Primary melanoma with liver metastasis ^3^						
5	M	71	Trunk	SSM	2.3	No
6	M	40	Extremities	NM	3.0	No
7	M	69	Trunk	SSM/NM	8.0	Yes
8	M	63	Trunk	SSM	2.2	No
9	F	71	Trunk	NM	7.0	Yes

^1^ F: female, M: male; ^2^ SSM: superficial spreading melanoma, NM: nodular melanoma; ^3^ Patients with at least a 5 year follow-up period.

## Data Availability

Not applicable.

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
