# Peer review of "Gene Expression Changes in Cytokine and Chemokine Receptors in Association with Melanoma Liver Metastasis"

_ijms, 2023, doi:10.3390/ijms24108901_

Round 1
Reviewer 1 Report
The manuscript presents an insightful analysis of cytokine/chemokines-receptor axis during in vitro invasion of melanoma cells when co-cultured with human hepatic sinusoidal endothelial cells (HHSECs). While the experimental design features prominently and analysis are well executed, there are several issues that need to be further addressed before the manuscript can be considered for publication.
Major issues
1. The data presented in the manuscript do not support the conclusion that IL11RA upregulation may play a key role in the formation of melanoma liver metastasis. The three melanoma cell lines (WM1366, WM278 and WM793B) that show non-invasive properties, regardless of the conditional medium from HHSECs, indicates that the gene expression in Figure.2A(a) is not triggered by HHSECs. Therefore, IL11RA should not be selected to show the commonality with the gene expression alterations in cell lines with increased invasiveness Figure.2A(b).
2. The manuscript lacks sufficient evidence to demonstrate that the findings specific to liver metastasis of melanoma. Although the authors investigated the cytokine/chemokine-receptor axis using HHSECs, there is no comparison with other tissue origins.
Minor issues
1. (Line 93) It would be appreciated if the authors could include pictures of cell invasion in the supplemental materials. The three melanoma cell lines (WM1366, WM278 and WM793B) that did not change their invasive properties coincidentally showed a high baseline of cell invasion, which might account for the observation of non-invasive capacity.
2. (Line 51-54) Please confirm if it is related. Blocking the axis of cytokine/chemokines and receptor may show benefit at the context of this manuscript.
3. (Line 74-75) Please clarify the difference between Liver sinusoidal endothelial cells (LSECs) and human hepatic sinusoidal endothelial cells (HHSECs), which was used in this study.
4. (Line 166-168) CCL20 showed statistically correlated with the invasive capacity of melanoma cells. However, the upregulation was only seen in one of the three cells with invasive capacity.
Minor editing of English language required.
Author Response
Dear Reviewer,
We would like to thank for your constructive criticism and the extensive effort you invested in providing your comments. We have carefully considered your requests and suggestions, and have made the necessary clarifications and changes in the revised version of the manuscript. In response to your comments we have rewritten certain sections of the manuscript and corrected the Figure legends. Furthermore, we have included supplementary figure (Supplementary Figure S1) depicting the invasion experiment.
We are confident that the revised version of our manuscript meets the high standards set by the International Journal of Molecular Sciences. We believe that the revisions we have made address the issues raised during the review process and further strengthen the quality and impact of our research. We excitedly anticipate the opportunity to submit this improved version for your consideration.
We have addressed the comments and questions point by point to improve clarity. In order to enhance readability, we have presented the original notes and questions in bold italics, followed by our corresponding answers in regular text. Please note that any new text is highlighted in red, while any deleted portions are shown as strikethrough text.
Enclosed please find our response as attachment.

Reviewer 2 Report
The study by Koroknai et al is well designed study, where they showed the close relationship between IL11RA overexpression in melanoma and liver metastasis. However, there are some major issues
1. From the abstract and throughout the manuscript it is not clear that why IL11RA has been emphasized
2. Most of the cited references are too old, it should be updated if literature is available
3. The meaning of * must be explained in the captions clearly.
4. Fig 4 should be formatted for overlapping text
Author Response
Dear Reviewer,
We would like to thank for your constructive criticism and the extensive effort you invested in providing your comments. We have carefully considered your requests and suggestions, and have made the necessary clarifications and changes in the revised version of the manuscript. In response to your comments we have rewritten certain sections of the manuscript and corrected the Figure legends. Furthermore, we have included supplementary figure (Supplementary Figure S1) depicting the invasion experiment.
We are confident that the revised version of our manuscript meets the high standards set by the International Journal of Molecular Sciences. We believe that the revisions we have made address the issues raised during the review process and further strengthen the quality and impact of our research. We excitedly anticipate the opportunity to submit this improved version for your consideration.
We have addressed the comments and questions point by point to improve clarity. In order to enhance readability, we have presented the original notes and questions in bold italics, followed by our corresponding answers in regular text. Please note that any new text is highlighted in red, while any deleted portions are shown as strikethrough text.
Please find our response as attachment.

Round 2
Reviewer 1 Report
Author’s diligent efforts in refining the content and enhancing the writing have significantly elevated the overall quality of the manuscript, allowing for a more engaging and accessible reading experience. I genuinely appreciate the improvement authors have made.
Accept in present form
Author Response
Thank you for your positive comment.
Reviewer 2 Report
The revised manuscript can be accepted.
Author should add bars in the supplementary fig if it can be done
Author Response
Thnak you for the suggestion, we have added bars to the images, and uploaded the new supplementary Figure 1.
Thanks again.